# Potential Utility of Biased GPCR Signaling for Treatment of Psychiatric Disorders

**DOI:** 10.3390/ijms20133207

**Published:** 2019-06-29

**Authors:** Hidetoshi Komatsu, Mamoru Fukuchi, Yugo Habata

**Affiliations:** 1Medical Affairs, Kyowa Pharmaceutical Industry Co., Ltd. (A Lupin Group Company), Osaka 530-0005, Japan; 2Department of Biological Science, Graduate School of Science, Nagoya University, Nagoya City 464-8602, Japan; 3Laboratory of Molecular Neuroscience, Faculty of Pharmacy, Takasaki University of Health and Welfare, Gunma 370-0033, Japan; 4Department of Food & Nutrition, Yamanashi Gakuin Junior College, Kofu 400-8575, Japan

**Keywords:** GPCR, biased ligand, dopamine D2 receptor, aripiprazole, quetiapine, β-arrestin, psychiatric disorder, schizophrenia, bipolar disorder

## Abstract

Tremendous advances have been made recently in the identification of genes and signaling pathways associated with the risks for psychiatric disorders such as schizophrenia and bipolar disorder. However, there has been a marked reduction in the pipeline for the development of new psychiatric drugs worldwide, mainly due to the complex causes that underlie these disorders. G-protein coupled receptors (GPCRs) are the most common targets of antipsychotics such as quetiapine and aripiprazole, and play pivotal roles in controlling brain function by regulating multiple downstream signaling pathways. Progress in our understanding of GPCR signaling has opened new possibilities for selective drug development. A key finding has been provided by the concept of biased ligands, which modulate some, but not all, of a given receptor’s downstream signaling pathways. Application of this concept raises the possibility that the biased ligands can provide therapeutically desirable outcomes with fewer side effects. Instead, this application will require a detailed understanding of the mode of action of antipsychotics that drive distinct pharmacologies. We review our current understanding of the mechanistic bases for multiple signaling modes by antipsychotics and the potential of the biased modulators to treat mental disorders.

## 1. Introduction

Despite tremendous success identifying genetic risk factors for major psychiatric illnesses such as schizophrenia, bipolar disorder, and depression, the field of psychiatric disorders still lags far behind other therapeutic areas of medicine in that not even a single gene has been conclusively demonstrated to provoke any of their symptoms. This is in large part due to the complex genetic as well as environmental and epigenetic risk factors that underlie these diseases. Furthermore, the rational design of multitarget drugs such as multi-acting receptor targeted antipsychotics (MARTA) has encountered considerable challenges in optimizing multiple structure-activity relationships while maintaining drug-like properties. These factors have led to a dramatic worldwide decline in the discovery and development of new psychiatric drugs [1,2,3]. Thus, novel approaches are required to bypass the obstacles in the drug discovery, as the needs for developing efficacious and well-tolerated treatments remain unequivocal.

G-protein coupled receptors (GPCRs) [1], also known as seven-transmembrane domain proteins, are the most common targets of antipsychotics such as quetiapine, aripiprazole, and olanzapine. GPCRs constitute the largest receptor superfamily that play critical roles in regulating a variety of physiological responses and account for nearly 30% of the Food and Drug Administration (FDA)-approved drug targets [4]. GPCRs can be divided into two types [1], odorant/sensory and non-odorant receptors. Odorant/sensory receptors detect external stimuli such as light, odors, tastes, and pheromones. Non-odorant receptors are expressed throughout the whole body and respond to a variety of ligands. They mediate numerous physiological responses including hemostasis, reproduction, cardiac function, immune function, metabolism, and neurotransmission [4]. The non-odorant GPCR superfamily consists of 367 receptors in humans and 392 in mice where 343 are common in the two species [5]. A large proportion of non-odorant GPCRs, approximately one fourth of which are orphan receptors, are rich in the central nervous system (CNS), especially in the brain [5,6]. For instance, dopamine, serotonin, glutamate, and acetylcholine receptors, all of which are well-known neuropharmacological targets, are highly expressed in the brain. This suggests that these brain-specific receptors have great potential as therapeutic targets for CNS disorders.

Neurons communicate with each other via neurotransmitters through two mechanisms referred to as fast and slow synaptic transmission, which involve two distinct classes of receptors. Ionotropic receptors comprise ligand-gated ion channels that induce fast synaptic transmission. In contrast, metabotropic receptors constitute GPCRs that cause slow synaptic transmission through intracellular signal transduction as well as induction of gene expression to exert antipsychotic actions [7,8,9]. Intriguingly, most neuropharmacological drugs are known to regulate GPCR activity in the central nervous system (CNS) [10].

GPCR activation elicits downstream G protein-dependent signaling followed by phosphorylation of the receptor by G protein-coupled receptor kinases (GRKs) [11]. The phosphorylation augments interaction of the receptor with β-arrestins, which in turn mediates desensitization of G-protein signaling and internalization of GPCRs [12,13,14,15]. It is now firmly established that GPCRs can act through multiple transducers, including canonical G-protein pathways and noncanonical β-arrestin-dependent pathways, to scaffold various signaling molecules such as kinases and phosphatases [16,17,18]. Unraveling of these distinct G-protein and β-arrestin signal transduction pathways has provided support for the concept of biased signaling, in which different types of ligands have the ability to stabilize distinct receptor conformations that in turn can elicit distinct signaling outcomes [19]. There are several physiologically relevant examples in which pharmacological agents can selectively target these different signaling pathways. For instance, a series of antipsychotics ranging from inverse to partial agonists and antagonists on D2 dopamine receptor (D2R)-mediated G-protein activation share the common property of potently antagonizing D2R/β-arrestin2 (βarr2) pathways, which suggests that a D2R-βarr2 antagonism is essential for antipsychotic efficacy [4]. In cardiovascular disorders such as hypertension, coronary artery disease and heart failure, β-blockers including alprenolol and carvedilol have been used for the treatment. These drugs not only antagonize G-protein-dependent pathways mediated by β-adrenergic receptors but also activate the receptor signaling pathways in a G protein-independent, β-arrestin-dependent fashion [20].

Accumulating evidence indicates that selective biased signaling may be extended to a wide variety of GPCRs currently targeted by pharmacological agents. The conventional agonists and antagonists are simultaneous stimulators and blockers of both G-protein-mediated signaling and β-arrestin-dependent signaling, respectively. The specificity of the ‘biased ligands’, which selectively target either G-protein-mediated signaling or β-arrestins-dependent signaling, is able to separate the benefits and the adverse effects of conventional drugs.

## 2. Canonical and Noncanonical GPCR Signaling Pathways

Conformational changes of GPCRs induced by their endogenous ligands lead to regulation of a plethora of cellular processes though both canonical G-protein pathways and noncanonical β-arrestin and G protein-coupled receptor kinase (GRK) pathways (Figure 1A). In G protein pathways, GPCRs need to couple with intracellular heterotrimeric G proteins, which are formed by Gα, Gβ and Gγ subunits. The Gα subunits can be classified into four major families (Gαs, Gαi/o, Gαq/11 and Gα12/13), each of which regulates key effectors and induces the generation of second messengers such as cAMP, Ca^2+^, and Inositol 1,4,5-triphosphate (IP3). These second messengers in turn trigger distinct signaling cascades. Notably, multiple distinct GPCRs can couple with the same Gα protein and the same receptor can also couple with more than one Gα protein. Gβγ subunits have both regulatory and signaling functions, serving either as scaffolds for kinases such as phosphatidylinositol 3-kinase (PI3K) and protein kinase D (PKD) or as modulators of ion channels (Figure 1A) [5,6]. Comprehensive identification of other GPCR interactors has further expanded the complexity of potential consequences of receptor signaling. The most studied receptor interactors are arrestins, which can negatively regulate the signaling by decoupling the activated receptors from G proteins and evoking their receptor internalization and endocytosis. However, they can also function as scaffolds to trigger additional signaling including activation of various mitogen-activated protein kinases (MPAKs), such as extracellular signal-regulated kinase (ERK) and c-Jun N terminal kinase (JNK) [5]. This is often termed the arrestin-dependent, G protein-independent signaling pathway, although the extent to which such signaling can be regulated by G protein-dependent signaling remains elusive. For instance, recent studies have revealed that G proteins, but not arrestins, are essential for the initiation of GPCR-mediated mitogen-activated protein kinase (MAPK) signaling, a fundamental signaling pathway that controls synaptic plasticity, proliferation, differentiation, and cell survival, whereas arrestin-dependent GPCR internalization can be achieved in the absence of active G proteins [7,8,9]. Recently, bioluminescence resonance energy transfer (BRET)-based and fluorescence resonance energy transfer (FRET) based biosensors have been extensively used to interrogate protein-protein interactions between receptors, G proteins, β-arrestins, and their numerous binding partners in living cells. This allows us to dramatically accelerate generation of novel biased drugs [10].

## 3. Modes of Action of Antipsychotics in Psychiatric Disorders

Dopamine is a catecholamine neurotransmitter that is deeply involved in CNS disorders such as schizophrenia, bipolar disorder, Parkinson’s disease, attention deficit hyperactivity disorder, and obsessive–compulsive disorder [7,11,12,13,14]. The observations led by Seeman and Snyder that antipsychotic drugs bind to D2Rs and psychostimulants that increase brain dopamine exacerbate psychotic symptoms strengthened the hypothesis of a hyperdopaminergic state of dopamine in schizophrenia [15,16,17]. Dopamine exerts its actions through GPCRs that are classified into two major subclasses of receptors, the D1 class including D1 receptor (D1R) and D5 receptor and the D2 class including D2R, D3 receptor, and D4 receptor [18], based on their ability to activate the stimulatory G protein Gs/olf or inhibitory G protein Gi/o signaling pathway, respectively. The striatum, in which D1R and D2R are most abundant [19], is the major input region of the basal ganglia and is strongly innervated by the ventral tegmental area (VTA) and the substantia nigra pars compacta (SNc). VTA is the origin of the dopaminergic mesolimbic pathway that is believed to be hyperactive in positive symptoms of schizophrenia [21,22]. The striatum consists of about 95% GABAergic medium-sized spiny neurons (MSNs) and 5% of interneurons including large aspiny cholinergic neurons [20,23,24]. MSNs is comprised of the striatonigral (direct) and striatopallidal (indirect) pathways and can be divided into two types of neurons based on their projections and the receptors and neuropeptides expressed [25]. The striatonigral neurons project onto the substantia nigra pars reticulata (SNr) and the medial globus pallidus (MGP) and specifically express neuropeptides substance P and D1R [26,27]. The striatopallidal neurons project onto the lateral globus pallidus (LGP). This pathway reaches the SNr/MGP through the subthalamic nucleus (STN). The striatopallidal neurons specifically express neuropeptide enkephalin, D2R, adenosine A2a receptor (Figure 1B), and two orphan GPCRs, GPR6 and GPR52 [26,27,28,29,30,31] (Figure 2). Both types of MSNs express orphan GPCR, GPR88 [28,32]. The direct and indirect pathways have opposite but balancing roles in regulating motor behavior [33]. The direct pathway facilitates locomotion whereas the indirect pathway abrogates movement [34]. MSNs are also deeply involved in addiction, motivation and reward, as well as in the manifestation of Parkinson’s disease and schizophrenia [19,21], however, their differential functions still remain elusive.

In the striatopallidal neurons, D2R activation evokes Gi/o-mediated signaling to inhibit intracellular cAMP accumulation and the protein kinase A (PKA)/dopamine and cAMP-regulated phosphoprotein of 32 kDa (DARPP-32) pathway, which can mediate numerous behavioral effects of dopamine [7,35,36] (Figure 1B). Gs-coupled GPR6, GPR52, and A2a, and Gi-coupled GPR88 also can regulate this pathway [28,31,32,37,38] (Figure 2). These Gs-coupled GPCRs are well known to potentiate NMDA (N-methyl-D-aspartate) receptor activity through phosphorylation of its receptor via cAMP/PKA (Figure 2) [39,40]. However, based on the initial finding that the dopamine-dependent locomotory response to amphetamine is remarkably attenuated in mice lacking βarr2 [41], Beaulieu JM, et al. showed that a βarr2-dependent signaling pathway downstream of D2R, which is distinct from Gi/o signaling, also participates in regulation of dopamine response [42]. They demonstrated that the D2R-βarr2 signaling can inhibit protein kinase B (PKB or AKT) activity, activate glycogen synthase kinase 3 beta (GSK3β), and mediate specific dopamine-dependent behaviors (Figure 1B) [43,44,45]. Intriguingly, a protein-protein interaction between D2R and DISC1 (disrupted in schizophrenia 1) was identified as a candidate initiator of hyperactive behaviors in mice that require βarr2 signaling to regulate AKT phosphorylation and subsequent GSK3β activation [46]. The mood stabilizer lithium inhibits GSK3β activation and interaction between βarr2 and AKT (Figure 1B) [45].

Antagonism of D2Rs in basal ganglia is the primary mode of action of antipsychotics, the first-line treatment for schizophrenia and bipolar disorder that share high levels of polygenic and pleiotropic molecular architecture [1,15,47]. Indeed, a genome wide association study (GWAS) identified the D2R gene within a schizophrenia-associated locus, further indicating the pivotal role of its signaling in the disease pathogenesis and treatment options [48]. However, antipsychotics are not clinically effective at alleviating cortical-related symptoms, such as negative symptoms and cognitive impairment [49]. Antipsychotics can be classified into two categories, typical and atypical antipsychotics (Figure 3A,B). Typical antipsychotics (first-generation antipsychotics) such as haloperidol mainly have an antagonistic activity for D2Rs, while atypical antipsychotics (second-generation antipsychotics) such as risperidone, quetiapine and clozapine possess antagonistic activities for serotonin 2A receptor (5-HT2A), D2Rs and other multiple GPCRs. Atypical antipsychotics are less likely to induce some side effects such as hyperprolactinaemia and extrapyramidal symptoms (EPS) than typical antipsychotics [50]. Although EPS induced by antipsychotics result from their excessive D2R binding in striatal regions, it is thought that therapeutic effectiveness also needs striatal D2R binding at a faster dissociation rate [51]. However, atypical antipsychotics possess their own characteristic side effects, which include weight gain, hypotension, and agranulocytosis.

Several findings argued against hyperdopaminergia as the singular cause of manifestation of schizophrenia [52], and additional evidence led to a revision of the dopamine hypothesis in which not only hyperdopaminergia in basal ganglia but also hypodopaminergia in frontal cortex are involved [53,54]. Based on this updated dopamine hypothesis, it was believed that all antipsychotics that block D2Rs would reverse only striatal hyperdopaminergia but not cortical hypodopaminergia. Thus, one would need to devise an antipsychotic drug that can simultaneously inhibit and activate dopamine signaling dependent on the brain regions. The findings that D2R can signal not only through canonical G protein pathways but also through noncanonical pathways that promote the formation of a signaling complex composed of 5AKT, PP2A, GSK3β, and βarr2 may open new avenues for pharmacologically targeting D2Rs for antipsychotic drug therapy [42,55]. The observation that β-blockers alprenolol and carvedilol stimulate signaling pathways in a G protein-independent and arrestin-dependent manner paved the way to explore this concept [56]. Conditional knockout mice lacking βarr2 or GSK3β in D2R-expressing striatal neurons (striatopallidal MSNs), but not in striatonigral MSNs, show disruption of the signaling scaffold complex and reduction of dopamine-dependent locomotor behavior, which mimics the mode of actions of antipsychotic drugs [45,57]. Haloperidol induces phosphorylation of AKT in mouse brains that could compensate for an impaired function of βarr2-GSK3β pathways in schizophrenia [58]. In vitro assays, all clinically effective antipsychotic drugs block D2R-βarr2 recruitment [4]. D2R-Gi/o-mediated pathways via DARPP-32, but not βarr2-GSK3β pathways, exacerbates haloperidol-induced catalepsy [36,45]. EPS and several side effects induced by antipsychotics are thought to be associated with Gi/o signaling [36,45]. Antipsychotics that selectively target the D2R-βarr2 pathway could be therapeutically beneficial with less adverse events such as EPS. In support of this hypothesis, BRD5814, a βarr2 biased D2R antagonist achieves efficacy in dopamine-induced hyperlocomotion with significantly reduced motoric side effects in mice [59]. In the following sections, modes of action of aripiprazole and quetiapine, commonly prescribed antipsychotics with unique characteristics, are updated.

### 3.1. Aripiprazole

Aripiprazole is a third-generation antipsychotic drug initially approved for the treatment of schizophrenia but found to be effective in bipolar disorder and major depressive disorder [60,61]. Aripiprazole acts as a partial agonist at D2R-Gi/o pathway but retains most of the properties of other atypical antipsychotics, such as 5-HT2A receptor antagonism. Under high dopaminergic tone, aripiprazole acts as a partial D2R-Gi/o antagonist, suggesting that it can normalize both hyperdopaminergic and hypodopaminergic states in D2R-Gi/o pathways [4,62,63]. Strikingly, aripiprazole, which displays agonistic activity on the Gi/o pathway but has no intrinsic agonistic activity on βarr2 recruitment, is able to fully block D2R-βarr2 translocation induced by dopamine (Figure 3C) [4]. Aripiprazole has been shown to lack the ability to induce internalization of D2R [64], a mechanism known to be βarr2 dependent [4]. These findings indicate that aripiprazole is a functionally selective (or biased) agent for D2R. On the other hand, haloperidol, clozapine, chlorpromazine, quetiapine, olanzapine, risperidone, and ziprasidone all potently antagonize both D2R-mediated Gi/o and βarr2 pathways from low to high dopaminergic tones (Figure 3B) [4]. Aripiprazole has few of the typical adverse events (AEs) of other antipsychotics, such as EPS, hyperprolactinemia, weight gain, metabolic disorders, and sedation [65]. This could be attributed to its unique biased activity and/or partial agonistic/antagonistic actions on D2Rs. Unfortunately, however, aripiprazole has in large measure failed to ameliorate the cognitive impairment in schizophrenia. Postmortem brain analyses of patients with schizophrenia have shown that mRNA transcripts of GABA synthesizing enzyme glutamic acid decarboxylase 67 (GAD67) and the calcium-binding protein parvalbumin, which are two markers of GABAergic fast-spiking interneurons (FSIs), are downregulated [66,67,68,69,70]. It is believed that disrupted cortical gamma rhythms in prefrontal cortex (PFC) result in cognitive impairment due to disruption of the FSI function in schizophrenia [71,72,73,74], which can be restored by D2R-βarr2 activation in cortical FSIs [57]. In PFC, D2Rs are expressed in GABAergic FSIs that regulate action potentials of glutamatergic pyramidal neurons. Although aripiprazole behaves as a partial D2R-βarr2 agonist only in the presence of GPCR kinase 2 (GRK2) overexpression in vitro [57], it exhibits a weak increase in action potential firing in cortical FSIs in mice [57]. This perhaps suggests why aripiprazole might not have been successful in reversing cortical-related symptoms in patients with schizophrenia. Taken all together, we propose that aripiprazole fully antagonizes D2R-βarr2 pathway while normalizing D2R-Gi/o pathway in hyperdopaminergic striatum of the disorder.

### 3.2. Quetiapine

Quetiapine, an atypical antipsychotic drug antagonizing multiple GPCRs including D2R and 5-HT2A [75,76], has been approved worldwide for various mental disorders, such as schizophrenia and bipolar disorder. In particular, quetiapine is one of the first-line antipsychotics for nearly all phases of bipolar disorder across treatment guidelines [47,77]. The antidepressant activity of quetiapine may be mediated, at least in part, by its metabolite norquetiapine through noradrenaline transporter inhibition [78]. Confirmatory studies of both quetiapine immediate-release (IR) and extended release (XR) tablets have demonstrated that quetiapine is effective and safe for bipolar disorder patients in a depressive state [79,80,81,82,83]. Increasing evidence reveals that patients with schizophrenia have abnormal expression of cytokines, suggesting that inflammation may be important in the pathogenesis of this disorder. Quetiapine is known to exhibit unique neuroprotective and anti-inflammatory properties [84,85,86]. In mice, quetiapine inhibits activation of astrocytes and microglia and reduce generation and release of two cytokines, tumor necrosis factor-α (TNF-α) and monocyte chemoattractant protein-1 (MCP-1). Meanwhile, quetiapine also prevents the loss of the synaptic protein synaptophysin (SYP) and myelin basic protein (MBP). These findings suggest that quetiapine may inhibit neuroinflammatory response from glial cells and block injury by the released cytokines to neurons and oligodendrocytes [85]. Quetiapine increases synthesis of ATP in astrocytes and protects GABAergic neurons from aging-induced cell death, leading to amelioration of anxiety-like behaviors in mice [86]. Quetiapine also exerts its neuroprotective effects against amyloid toxicity by enhancing the release of brain-derived neurotrophic factor (BDNF) from cultured astrocytes [87]. It was reported that quetiapine exhibits unique temporal and regional regulation of epidermal growth factor receptor (EGFR)-ERK pathway and its downstream transcriptional targets, p90RSK and c-Fos in PFC and striatum in mice, where cortical ERK1 phosphorylation by aripiprazole is EGFR independent whereas striatal ERK1 activation by quetiapine is EGFR dependent (Figure 3D) [88]. This cannot be explained in part by partial 5-HT1A receptor activation by quetiapine or its metabolite N-Desalkylquetiapine. It is because their free plasma concentrations in human reach submicromolar levels at a maximum while approximately 5 to 10 micromolar concentrations of these two compounds are needed to elicit the receptor stimulation [78,89]. Intriguingly, quetiapine, clozapine, and olanzapine induce the activation of ERK and AKT possibly through 5-HT2A receptor activation whereas risperidone, aripiprazole, and typical antipsychotics fail to activate these kinases at the concentrations that can be reached during pharmacological therapy. These observations suggest that molecular mechanism of quetiapine action is complex and could involve biased signaling (Figure 3D) [90].

## 4. Elucidation of D2R-Mediated Biased Functions by Genetically Engineered Biased D2R Mutants

The function of D2Rs has been examined through the use of pharmacological agents and by genetic deletion approaches. Recent studies using cell type-specific knockouts illustrate the necessity of D2Rs for many behavioral and physiological functions [91,92,93]. Striatopallidal MSN-D2R knockout mice mimic whole-body D2R knockout mice in that they exhibit abnormalities in skilled movements, locomotion, and the acute response to cocaine [94]. More recently, an alternative strategy has been used to generate D2R mutants that activate only D2R-Gi/o or D2R-βarr2 pathway. Crystallographic and modeling studies on GPCRs have postulated the residues near the second intracellular loop and those in the third transmembrane domain to interact with the C terminus of the α subunit of G proteins and βarr2. The combinatorial mutations in the second intracellular loop A135R and M140D generated a βarr2-biased D2R, while L125N and Y133L generated a Gi/o-biased D2R [95]. Using these genetically engineered biased D2R mutants in mice in vivo, Rose et al. have revealed the distinct roles of D2R-mediated biased signaling pathways in MSNs [96]. Amphetamine- or cocaine-induced locomotion is significantly reduced in mice deleted for D2Rs in striatopallidal MSNs. These drugs are psychostimulants known to increases presynaptic dopamine release and locomotion. Intriguingly, these behavioral reductions can be partially or fully restored by expressing Gi/o-biased or βarr2-biased D2R mutant proteins in striatopallidal MSNs lacking wild type D2Rs, suggesting that both D2R-Gi/o and D2R-βarr2 pathways participate in hyperdopaminergic behaviors [96]. In contrast, phencyclidine (PCP)-induced locomotion is potentiated in mice expressing the βarr2-biased D2R mutant but not the Gi/o-biased D2R mutant in the striatopallidal MSNs, suggesting that NMDAR blockade-induced locomotion is mostly driven by D2R-βarr2 signaling [96].

Several studies have illustrated the importance of cAMP–PKA–DARPP32 signaling in striatopallidal MSNs for antipsychotic-induced phosphorylation of ribosomal protein S6 and histone H3, as well as the expression of c-fos and egr-1 [97,98]. On the other hand, ERK 1/2, which is activated by both D2R-Gi/o and D2R-βarr2 signaling [95], is critical for antipsychotic-induced signaling as well [97,99,100]. The in vivo study using biased D2R mutants confirms that haloperidol-induced signaling is mostly mediated by D2R-Gi/o blockade [96]. This is consistent with the hypothesis that antipsychotics induce gene transcriptional changes and long-term effects through D2R-Gi/o signaling disruption [36,101], whereas D2R-βarr2 blockade mediates non-transcriptional changes that lead to the improvement in the positive symptoms of schizophrenia [4,45,101]. This study also shows that antagonism of D2R-βarr2 signaling is capable of driving the gene expression of egr-1. Thus, an overlap of D2R-Gi/o and D2R-βarr2 signaling likely occurs. The recent study shows that the βarr2-biased D2R mutant can restore the locomotor activities in a manner indistinguishable from wildtype D2R whereas it fails to enhance motivation. This suggests that motivation requires D2R-Gi/o activation [102]. The approaches to reconstitute mice lacking D2Rs with these D2R mutants in various neuronal populations from different brain regions allow us to unravel the contributions of each pathway to a variety of dopamine-dependent behaviors.

## 5. Potential Biased Antipsychotics

Although atypical antipsychotics show beneficial effects on positive symptoms in schizophrenia with less side effects such as extrapyramidal symptoms, none of these drugs are expected to effectively improve negative symptoms and cognitive dysfunction. However, recent efforts support the idea that biased D2R ligands effectively restore these impairments. Allen et al. have discovered several functionally selective biased D2R-βarr2 ligands, UNC9975, UNC0006, and UNC9994, by exploring multiple regions of the aripiprazole template [101]. As described above, aripiprazole can act as a partial agonist at D2R-Gi/o and/or D2R-βarr2 pathways dependent on the cell types [64,103]. On the other hand, these biased compounds display D2R-βarr2 agonistic/antagonistic activity in various cellular assays, but show minimal D2R-Gi/o agonist activity [57,101]. These compounds exhibit inhibitory effects on amphetamine- or PCP-induced hyperlocomotion via βarr2-mediated pathway with less motoric side effect in mice. UNC9975 and UNC0006 induce catalepsy in βarr2-knockout mice, suggesting that they signal through β-arrestin2 in vivo and this signaling may protect against motoric side effects due to antagonism at D2R-Gi/o pathway [36,101]. Importantly, D2R-βarr2 agonistic activities of these biased D2R-βarr2 compounds as well as aripiprazole depend the expression level of GRK2. In vitro assay using HEK293T cells, exert their ability to activate the D2R-βarr2 pathway in GRK2-overexpressing cells. In contrast, these compounds can act as antagonists at D2R-βarr2 pathways in the cells with low levels of GRK2. GRK2 expression in the PFC is higher than that in striatum [57], suggesting that they act as D2R-βarr2 agonists in the PFC and simultaneously act as D2R-βarr2 antagonists in the striatopallidal MSNs [57]. Most recently, UNC9994 is found to exhibit antipsychotic-like activities through the heterodimerization of A2a receptors with D2Rs in striatopallidal MSNs [104]. Since A2a receptor knockout mice display less catalepsy in the haloperidol-induced model [105], D2R-βarr2 pathways are likely to mediate EPS.

A series of observations obtained by biased D2R ligands indicate that a functionally selective compound, which activates D2R-βarr2 pathway and reduces Gi activity in hypodopaminergic PFC while simultaneously blocking D2R-βarr2 pathways in hyperdopaminergic striatum, would exhibit beneficial effects on schizophrenic symptoms (Figure 4). It is intriguing how such a compound can affect short- and long-term neural plasticity. The regulation of D2R activity causes two types of cellular events, gene transcription-dependent and independent signal transduction mediating long-lasting and acute synaptic plasticity, respectively. The transcription-dependent event is mainly caused by reduction of D2R-Gi/o signaling, resulting in intracellular cAMP increase. This in turn activates not only PKA-dependent cellular events but also NMDAR-derived Ca2+/calcineurin signaling, eventually leading to altered expression of genes including those related to neuronal plasticity such as c-fos, egr-1, and Bdnf [106] (Figure 4A). Indeed, haloperidol administration is known to enhance BDNF transcripts in the striatum and hippocampus [107]. This transcription-dependent event would participate in long-lasting effects on cellular and behavior responses. Meanwhile, the nontranscriptional event, which is thought to be mediated by D2R-βarr2 signaling in striatopallidal MSNs, also can regulate the behavioral responses such as PCP-induced hyperlocomotion in mice [96], suggesting that this signaling may contribute to the execution of acute effects by antipsychotics. βarr2 serves as a complex component that plays a key role in D2R-mediated AKT signaling (Figure 1B). Although AKT controls GSK3β activity, it remains equivocal whether AKT/GSK3β signaling can be activated or inhibited by D2Rs [108]. It has been demonstrated that GSK3β signaling contributes to brain development including neurogenesis, neuronal polarization, and axonal outgrowth [109]. The imbalance of AKT and GSK3β activities is thought to be involved in manifestation of psychiatric disorders [110]. Thus, it is possible that biased D2R-βarr2 agonism in PFC may contribute to neurite outgrowth and can normalize neural network in these disorders (Figure 4A). UNC9994, but not aripiprazole, enhances firing of GABAergic FSIs in the mouse PFC in D2R-, GRK2, and βarr2-dependent manner, suggesting that D2R-βarr2 agonism can induce FSI excitability (Figure 4A). This excitability may be attributed to the relative balance between Gi/o and βarr2 pathways because D2R-βarr2 agonism (increased FSI firing) is counteracted by its reciprocal D2R-Gi/o agonism (inhibition of FSI firing). In an hypodopaminergic state of the illness, aripiprazole can act as a partial Gi/o agonist that leads to a decline in intracellular cAMP, whereas UNC9994 induces no Gi/o activity [101]. Previously, Fukuchi et al. have demonstrated that Gs-coupled GPCR-mediated cAMP/PKA pathway selectively activates NMDAR-derived Ca2+/calcineurin pathway, resulting in the efficient induction of brain-derived neurotrophic factor (BDNF) and other plasticity-related genes via cAMP-response element-binding protein (CREB)-regulated transcriptional coactivator 1 (CRTC1)/CREB-dependent transcriptional activation [106] (Figure 4A). A relative balance between Gs and Gi pathways over intracellular cAMP production regulates the transcription of a series of plasticity-related genes including BDNF. BDNF is a member of neurotrophins and exerts a variety of neural functions including memory consolidation [111]. Reduced BDNF level in PFC has been reported in the patients with schizophrenia [112,113]. Disruption of activity-dependent BDNF expression causes fewer cortical inhibitory neurotransmission [114]. Considering the observations that cortical GABAergic interneurons in schizophrenia are disrupted [71], low BDNF level is likely to be associated with fewer inhibitory neurotransmissions in schizophrenia. In support, low levels of BDNF and its receptor, TrkB, are significantly correlated with the reduction of GAD67 expression [113]. These findings suggest that a selective D2R-βarr2 ligand can act as an agonist in hypodopaminergic PFC, leading to amelioration of negative symptoms and cognitive dysfunction through the normalization of excitatory/inhibitory neurotransmissions in psychiatric disorders (Figure 4A). In contrast, a selective D2R-βarr2 ligand such as UNC9994 can act as an antagonist in the hyperdopaminergic state of striatopallidal MSNs (Figure 4B) [57]. This D2R-βarr2 antagonism is likely to elicit the AKT activation followed by the phosphorylation and inactivation of GSK3β in the MSNs, leading to the improvement of positive symptoms in schizophrenic patients (Figure 4B). In the striatopallidal MSNs, typical and atypical antipsychotics can antagonize both G protein and βarr2 pathways (Figure 3B), whereas a selective D2R-βarr2 ligand can specifically block the βarr2-dependent pathway, evoking less adverse events (AEs) such as EPS (Figure 4B). This is because haloperidol-induced catalepsy in mice is induced by the inhibition of D2R-Gi/o/DARPP-32 pathway in striatopallidal MSNs. In support of this, D2R-βarr2 ligands exhibit antipsychotic-like properties with less motoric side effect [59,101]. Taken all together, D2R-βarr2 biased ligands have the potential to normalize the levels of dopamine neurotransmission in the dopamine-excessive striatal region and dopamine-deficient cortical region simultaneously, and provide safer and more broadly effective therapies for mental disorders.

## 6. Conclusions

A tremendous number of studies with advanced technologies including next-generation sequencing in recent years have revealed that major psychiatric disorders, such as schizophrenia and bipolar disorder, share molecular and genetic architecture. However, not even a single gene that provokes any psychiatric symptoms has not been identified so far. GPCRs, especially D2R, are well known therapeutic targets for the disorders. A biased GPCR ligand has the potential to improve mental illness with fewer adverse effects.

## Figures and Tables

**Figure 1 ijms-20-03207-f001:**
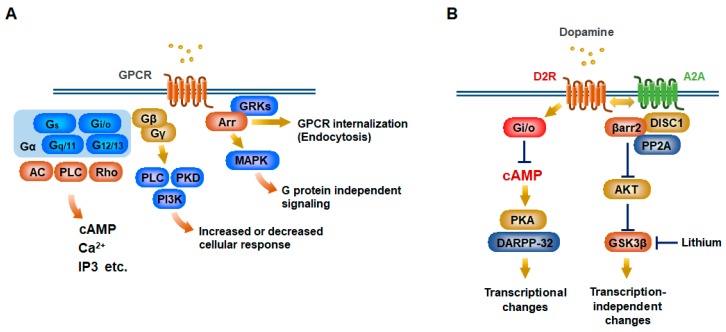
Schematic illustration of G protein-coupled receptor (GPCR) signaling. (**A**) Canonical G protein-coupled receptor (GPCR) signaling occurs through heterotrimeric G proteins (Gα, Gβ, and Gγ). Upon GPCR activation, the Gα and Gβγ subunits dissociate and can induce each downstream signaling. Gα proteins can be subdivided into four types of families with different signaling components including adenylyl cyclase (AC) for Gs and Gi, phospholipase C (PLC) for Gq, Rho for G12. Gβ and Gγ subunits further diversify into distinct signaling responses such as PLC, phosphatidylinositol 3-kinase (PI3K), and protein kinase D (PKD) pathways. In a noncanonical GPCR pathway, arrestins (Arr) bind to the activated receptor that has been phosphorylated by G protein-coupled receptor kinases (GRKs). This leads to desensitization of G protein signaling by the receptor internalization and endocytosis. Meanwhile, this signaling can elicit additional intracellular responses including mitogen-activated protein kinase (MAPK) activation. (**B**) D2 dopamine receptor (D2R) activation stimulates a Gi/o protein that decreases intracellular cAMP in striatopallidal medium-sized spiny neurons (MSNs). This negatively regulates protein kinase A (PKA)/dopamine and cAMP-regulated phosphoprotein of 32 kDa (DARPP-32) signaling, eventually leading to transcriptional changes. D2R activation also facilitates its interaction with adenosine A2a receptor (A2A) and DISC1 (disrupted in schizophrenia 1) and induces β-arrestin2 (βarr2) signaling via heterodimeric D2R/A2A, which inhibit protein kinase B (AKT) activity. AKT phosphorylates glycogen synthase kinase 3 beta (GSK3β) to prohibit its activation. Mood stabilizer lithium negatively regulates GSK3β activation and interferes with interaction between βarr2 and AKT.

**Figure 2 ijms-20-03207-f002:**
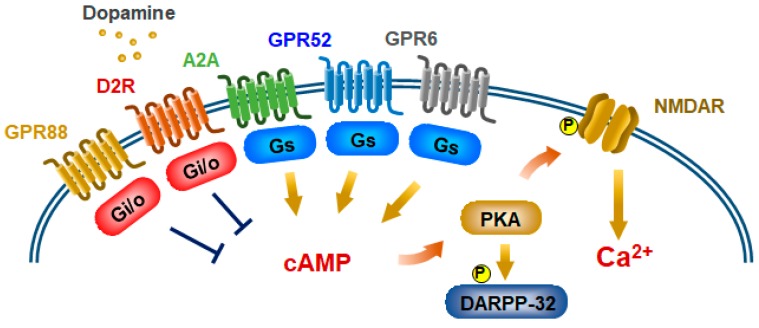
Model of GPCR-mediated NMDA receptor pathways in striatopallidal MSNs. Gs-coupled GPCRs potentiate NMDA receptor activity through phosphorylation of its receptor via cAMP/PKA. Gs-coupled GPR6, GPR52, and A2a receptors, and Gi/o-coupled GPR88 are expressed in striatopallidal MSNs. Activated Gi/o-coupled D2R and GPR88 inhibit NMDA receptor activity through inhibition of cAMP/PKA signaling.

**Figure 3 ijms-20-03207-f003:**
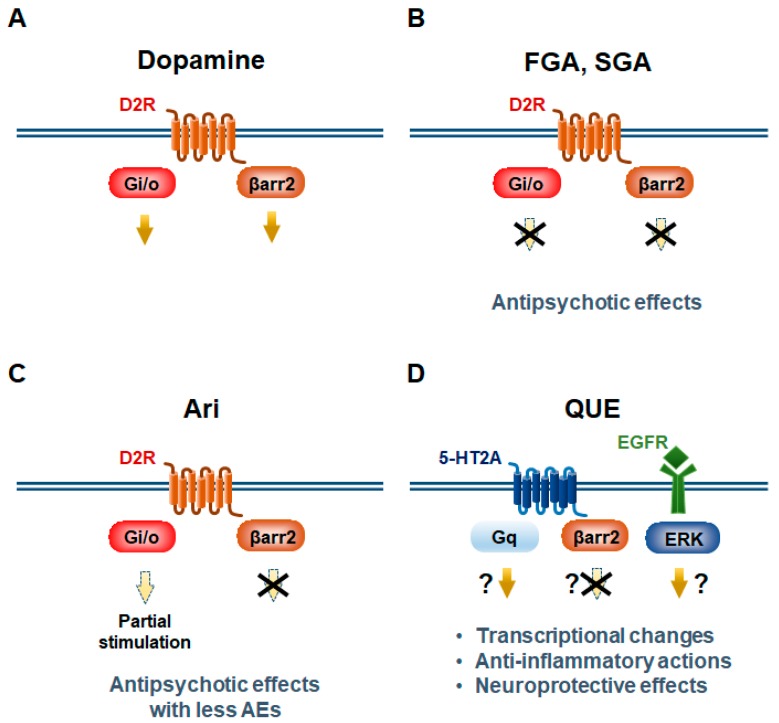
Proposed antipsychotic actions to confer bias of GPCR signaling. D2R-mediate Gi/o and β-arrestin2 (βarr2) pathways in striatopallidal MSNs (**A**–**C**). First-generation (FGA) and second-generation antipsychotics (SGA) antagonize both of these two pathways (**B**). Aripiprazole (Ari) functions as a partial agonist/antagonist at D2R-Gi/o signaling, whereas it fully antagonizes D2R-βarr2 signaling (**C**). Quetiapine (QUE) blocks 5-HT2A-βarr2 signaling and activates 5-HT2A-Gq signaling. Quetiapine also activates epidermal growth factor receptor (EGFR)-ERK pathway (**D**).

**Figure 4 ijms-20-03207-f004:**
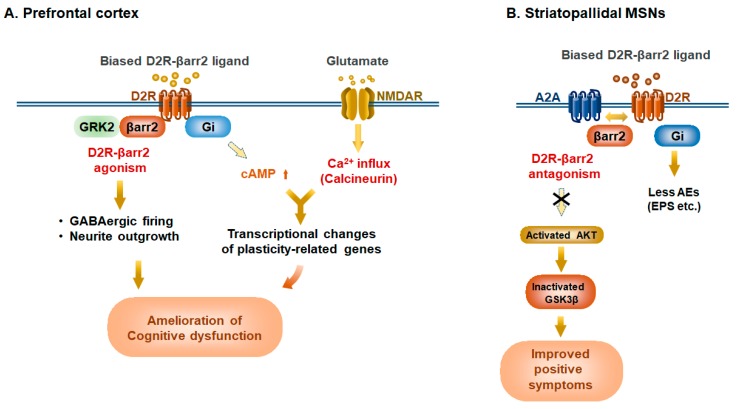
Potential therapeutic benefits of Selective D2R-β-arrestin2 ligands in prefrontal cortex and basal ganglia in schizophrenia. β-arrestin2 (βarr2)-biased ligands, such as UNC9994, can act as a D2R-βarr2 agonist in the hypodopaminergic prefrontal cortex but a D2R-βarr2 antagonist on hyperdopaminergic tone of striatopallidal MSNs. (**A**) In the hypodopaminergic state in the prefrontal cortex (PFC), D2R-βarr2 ligands can act as an agonist, which evokes action potential firing of cortical GABAergic fast-spiking interneurons (FSIs) and promotes neurite outgrowth through the AKT/GSK3β pathway. In parallel, the reduced Gi activity induces the rise of intracellular cAMP levels, resulting in the transcriptional changes of neuronal plasticity-related genes including egr-1, and Bdnf. This cAMP increase also potentiates NMDA receptor activity through cAMP/PKA, eventually leading to transcriptional changes. These events would participate in long-lasting effects on the neuronal circuit to ameliorate cognitive impairment in the disorder. (**B**) In the hyperdopaminergic state of striatopallidal MSNs, D2R-βarr2 ligands can act as antagonists, promoting heterodimerization of A2a receptor (A2A) with D2R and leading to the activation of AKT, which in turn inhibits GSK3β activation. Meanwhile, D2R-βarr2 ligands have no effect on the Gi/o pathway. These events would ameliorate positive symptoms of schizophrenia with less adverse events (AEs) such as extrapyramidal symptoms (EPS).

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
