# Peer review of "Potential Utility of Biased GPCR Signaling for Treatment of Psychiatric Disorders"

_ijms, 2019, doi:10.3390/ijms20133207_

Reviewer 1 Report

This is a generally well-written review. My comments are below:

Line 70: “It is now firmly established that GPCRs can signal through multiple transducers, including canonical G-protein pathways and noncanonical β-arrestin-dependent pathways, to scaffold various signaling molecules  such as kinases and phosphatases.” – this sentence is hard to understand, I suggest to replace the first “signal” by “act”?

Line 74 – The concept of biased signalling is not that “each signalling has the ability to mediate distinct physiological responses” , this is indeed a potential consequence of biased signalling. The concept of biased signalling refers to the ability of different ligands to stabilize distinct receptor conformations, linked to distinct signalling outcomes.

Line 78 – References 20-22 do not refer to the bARR antagonism as a mechanism of action for some antipsychotics. In fact, I think there is an important mix up of references throughout the manuscript.

Line 88 – needs “, respectively” at the end of the sentence.

Figure 1 – there is substantial evidence that the AKT-GSK3b pathway does not require Adenosine receptors and can indeed be mediated by D2Rs alone.

Line 204 – “black” should read “block”?

Line 217 – Allen et al showed that antipsychotics are not bARR2 antagonists, but partial agonism at this pathway.

Line 256 – the intrinsic efficacy of a ligand (e.g aripiprazole) should not be dependent on GRK2 overexpression. GRK2 overexpression, allows for the detection of such partial agonism, but does not condition it.

Line 302 – Another recent study using a D2 biased mutant, Donthamasetti et al., Mol Phychiatry 2018, should be discussed.

General – the section on quetiapine seems a bit out of the scope of the manuscript, as there is a lack of support for it being a biased ligand.

Additionally, the authors are encouraged to discuss recent publications that suggest and important role on ligand-binding kinetics on the actions of biased ligands at the D2R as well as other antipsychotics.

Author Response

Line 70: “It is now firmly established that GPCRs can signal through multiple transducers, including canonical G-protein pathways and noncanonical β-arrestin-dependent pathways, to scaffold various signaling molecules such as kinases and phosphatases.” – this sentence is hard to understand, I suggest to replace the first “signal” by “act”?

Reply: we replaced it as you suggested. Please see the attachment.

Line 74 – The concept of biased signalling is not that “each signalling has the ability to mediate distinct physiological responses” , this is indeed a potential consequence of biased signalling. The concept of biased signalling refers to the ability of different ligands to stabilize distinct receptor conformations, linked to distinct signalling outcomes.

Reply: we made the correction as you pointed out. Please see the attachment.

Line 78 – References 20-22 do not refer to the bARR antagonism as a mechanism of action for some antipsychotics. In fact, I think there is an important mix up of references throughout the manuscript.

Reply: we made the correction to the appropriate references as you pointed out. Please see the attachment.

Line 88 – needs “, respectively” at the end of the sentence.

Reply: we added it as you pointed out. Please see the attachment.

Figure 1 – there is substantial evidence that the AKT-GSK3b pathway does not require Adenosine receptors and can indeed be mediated by D2Rs alone.

Reply: this point remains elusive. Investigation by selective biased D2R-βarr2 ligands is likely to reveal this point. This figure is to show that A2A-D2R interaction can regulate beta-arrestin activity, not to show that A2A can regulate the AKT-GSK3b.

Line 204 – “black” should read “block”?

Reply: we made the correction as you pointed out. Please see the attachment.

Line 217 – Allen et al showed that antipsychotics are not bARR2 antagonists, but partial agonism at this pathway.

Reply: we made the correction to the appropriate reference at this line. In some conditions in vitro, antipsychotics look bARR2 antagonists. Please see the attachment.

Line 256 – the intrinsic efficacy of a ligand (e.g aripiprazole) should not be dependent on GRK2 overexpression. GRK2 overexpression, allows for the detection of such partial agonism, but does not condition it.

Reply: we made the correction and add the appropriate reference at this line. Please see the attachment.

Line 302 – Another recent study using a D2 biased mutant, Donthamasetti et al., Mol Phychiatry 2018, should be discussed.

Reply: we appreciate this comment. This study was discussed from line 334 as you pointed out. Please see the attachment.

General – the section on quetiapine seems a bit out of the scope of the manuscript, as there is a lack of support for it being a biased ligand.

Reply: it is because quetiapine is one of the most prescribed antipsychotics and have the potential biased activity as described in the section.

Additionally, the authors are encouraged to discuss recent publications that suggest and important role on ligand-binding kinetics on the actions of biased ligands at the D2R as well as other antipsychotics.

Reply: we appreciate this comment. It is still difficult to understand and predict the biased actions of antipsychotics from the aspects of their drug-binding kinetics. We would like to discuss this in the future.

Reviewer 2 Report

This is a very well written and updated review on the potential utility of biased GPCR dependent signaling for therapeutical strategies of psychiatric disorders.

The introduction part is well written, focused on the role of GPCRs in brain functions. Then the authors review the G-protein dependent and independent signaling pathways, relevant also in brain functions. This part is very detailed which is very good for the general audience. Further, the authors describe the mode of action of antipsychotics, revealing the central role of D2R in the generation of antipsychotics.

Lastly, the authors describe two antipsychotics, aripiprazole, and quetiapine.  It would be very nice to have the pharmacological properties (Ki values for other receptors) displayed in tables (e.g table 1, the pharmacological profile of aripiprazole and table 2 for quetiapine).

Overall it is an important review, well written and updated. 

Author Response

Lastly, the authors describe two antipsychotics, aripiprazole, and quetiapine.  It would be very nice to have the pharmacological properties (Ki values for other receptors) displayed in tables (e.g table 1, the pharmacological profile of aripiprazole and table 2 for quetiapine).

Reply: we really appreciate this comment. The pharmacological Ki properties of antipsychotics have been shown in many publications and widely prevailed. We like to omit this.